# Selected Flavonols Targeting Cell Death Pathways in Cancer Therapy: The Latest Achievements in Research on Apoptosis, Autophagy, Necroptosis, Pyroptosis, Ferroptosis, and Cuproptosis

**DOI:** 10.3390/nu16081201

**Published:** 2024-04-18

**Authors:** Dominika Wendlocha, Robert Kubina, Kamil Krzykawski, Aleksandra Mielczarek-Palacz

**Affiliations:** 1Department of Immunology and Serology, Faculty of Pharmaceutical Sciences in Sosnowiec, Medical University of Silesia in Katowice, 41-200 Sosnowiec, Poland; apalacz@sum.edu.pl; 2Silesia LabMed: Centre for Research and Implementation, Medical University of Silesia in Katowice, 41-752 Katowice, Poland; rkubina@sum.edu.pl (R.K.); kamil.krzykawski@sum.edu.pl (K.K.); 3Department of Pathology, Faculty of Pharmaceutical Sciences in Sosnowiec, Medical University of Silesia in Katowice, 41-200 Sosnowiec, Poland

**Keywords:** cancer, flavonol, cell death, apoptosis, autophagy, pyroptosis, cuproptosis, ferroptosis

## Abstract

The complex and multi-stage processes of carcinogenesis are accompanied by a number of phenomena related to the potential involvement of various chemopreventive factors, which include, among others, compounds of natural origin such as flavonols. The use of flavonols is not only promising but also a recognized strategy for cancer treatment. The chemopreventive impact of flavonols on cancer arises from their ability to act as antioxidants, impede proliferation, promote cell death, inhibit angiogenesis, and regulate the immune system through involvement in diverse forms of cellular death. So far, the molecular mechanisms underlying the regulation of apoptosis, autophagy, necroptosis, pyroptosis, ferroptosis, and cuproptosis occurring with the participation of flavonols have remained incompletely elucidated, and the results of the studies carried out so far are ambiguous. For this reason, one of the therapeutic goals is to initiate the death of altered cells through the use of quercetin, kaempferol, myricetin, isorhamnetin, galangin, fisetin, and morin. This article offers an extensive overview of recent research on these compounds, focusing particularly on their role in combating cancer and elucidating the molecular mechanisms governing apoptosis, autophagy, necroptosis, pyroptosis, ferroptosis, and cuproptosis. Assessment of the mechanisms underlying the anticancer effects of compounds in therapy targeting various types of cell death pathways may prove useful in developing new therapeutic regimens and counteracting resistance to previously used treatments.

## 1. Introduction

Based on the GLOBOCAN 2022 data on cancer incidence and mortality released by the International Agency for Research on Cancer, an estimated 19.98 million new cases worldwide and nearly 9.7 million deaths were reported in 2022. The main cancers include breast, lung, colon, prostate, and stomach cancer. According to estimates, by 2040, the incidence of cancer will increase to 47% and amount to 28.4 million [1]. Currently available treatments include a comprehensive, interdisciplinary, and holistic approach to cancer therapy. The latest ones focus on both the use of groundbreaking therapies and the development of new anticancer treatments and chemopreventive strategies. Chemoprevention can be developed at various levels, taking into account the mechanisms of influence that take place at individual stages of carcinogenesis, including both the initiation and progression stages. Complex and multi-stage processes are accompanied by a number of phenomena related to the potential involvement of various chemopreventive factors including compounds of natural origin such as flavonols. Their use is not only promising but also a recognized strategy for cancer treatment.

Flavonols represent the most prevalent flavonoids distributed throughout the entire plant kingdom. They have an unsaturated C ring at the C2-C3 position, which is usually connected to a hydroxyl group at the C3 position and an oxygen group at the C4 position. The positions of the hydroxyl group are responsible for the biological activity of the compounds. The main flavonols include quercetin (QUE), kaempferol (KEM), myricetin (MYR), isorhamnetin (ISO), galangin (GAL), fisetin (FIS), and morin (MOR) (Table 1) [2,3]. In reference to the multi-stage model of carcinogenesis, there are many chemopreventive strategies involving these compounds, which may be related to the regulation of mechanisms of various types of cell death.

Cell death plays a key role in maintaining homeostasis by removing damaged cells; moreover, it may also constitute a pathological response to harmful stimuli. The Cell Death Nomenclature Committee has developed a set of guidelines for dividing the modes of cell death into accidental cell death (ACD) and regulated cell death (RCD) [4]. RCD includes apoptosis, autophagy, necroptosis, ferroptosis, pyroptosis, NETosis parthanatos, entosis, lysosome-dependent cell death, alkaliptosis, and oxeiptosis [5].

Several studies suggest that within cancer cells, the basic functions of apoptosis are disturbed, probably due to numerous genetic defects. The possibility of the occurrence of phenotypes resistant to the induction of apoptosis and other types of cell death is emphasized. Thus, cells resistant to apoptosis are also insensitive to the effects of some therapies dedicated to a given type of cancer. Accordingly, over the last decades, cancer therapy research has focused on the development of improved pharmacotherapy and radiotherapies aimed at improving the sensitivity of cells to death and the resulting reduction in tumor volume [6]. The latest research also indicates the important role of newly discovered types of RCD, which include cuproptosis, pyroptosis, necroptosis, and ferroptosis. Mammalian cells, when subjected to irreversible disruptions in their intracellular or extracellular microenvironment, may trigger various signal transduction cascades, ultimately resulting in cell demise. Each of these RCD patterns is initiated and transmitted by molecular mechanisms that exhibit a significant degree of interconnection [3]. The basic differences in the mechanisms of the types of cell death described in this paper are presented in Figure 1.

**Table 1 nutrients-16-01201-t001:** List of flavonols involved in the regulation of different types of cell death [7,8,9,10,11,12,13,14].

Flavonoid	Chemical Formula	Structure	Cell Death
Quercetin	C_15_H_10_O_7_	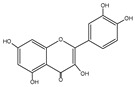	→ Apoptosis → Autophagy → Ferroptosis → Necroptosis
Kaempferol	C_15_H_10_O_6_	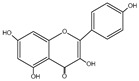	→ Apoptosis → Autophagy → Pyroptosis → Ferroptosis → Necroptosis
Galangin	C_15_H_10_O_5_	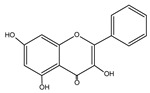	→ Apoptosis → Autophagy → Pyroptosis → Ferroptosis
Myricetin	C_15_H_10_O_8_	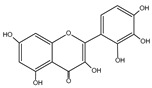	→ Apoptosis → Autophagy → Pyroptosis
Isorhamnetin	C_16_H_12_O_7_	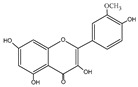	→ Apoptosis → Autophagy
Fisetin	C_15_H_10_O_6_	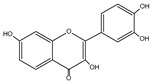	→ Apoptosis → Autophagy → Pyroptosis → Necroptosis
Morin	C_15_H_10_O_7_	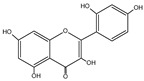	→ Apoptosis → Autophagy

## 2. Apoptosis

Apoptosis or Programmed Cell Death is tightly regulated and is necessary to maintain homeostasis of the body. It plays a key role in maintaining a number of physiological processes, such as embryonic development and adult tissue homeostasis, but is also a mechanism for limiting the growth of cancer cells [15]. The process of controlled cell death consists of three stages: activation of caspases, breakdown of DNA and proteins, and changes in the cell membrane and their recognition by phagocytic cells [16]. The molecular mechanism of apoptosis is a well-understood process and requires activation of caspase proteases through an intrinsic pathway or an extrinsic pathway by activation of death receptors such as Fas and DR4/5 through their death-inducing ligands, for example, FasL and TRAIL, respectively [17,18].

The extrinsic pathway associated with the activation of death receptors begins with the formation of the ligand–receptor complex. The most well-known death receptors are TNF receptor type 1 (TNFR1) and a related protein called Fas (CD95), along with their ligands TNF and Fas ligand (FasL). These receptors with an intracellular death domain recruit the adapter proteins TNF receptor-related death domain (TRADD), Fas-related death domain (FADD), and cysteine proteases. The binding of a death ligand to the death receptor causes the creation of a binding site for the adapter protein, and the entire ligand–receptor–adapter protein complex is known as the death-inducing signaling complex (DISC). DISC then initiates the assembly and activation of procaspase 8. The activated form of the enzyme, caspase 8, is an initiator caspase that initiates apoptosis by cleaving subsequent caspases [16,19].

In the case of the intrinsic mitochondrial pathway, activation occurs with the participation of internal stimuli, such as genetic defects, hypoxia, an increased cytosolic Ca^2+^ concentration, and oxidative stress. Activation of the mitochondrial pathway leads to the release of pro-apoptotic molecules such as cytochrome-c from the mitochondria into the cytoplasm. This pathway is strictly regulated by the so-called Bcl-2 family proteins, which include pro-apoptotic proteins such as Bax, Bak, Bad, Bcl-Xs, Bid, and anti-apoptotic proteins such as Bcl-2 and Bcl-X. Other apoptotic factors released from the mitochondrial intermembrane space into the cytoplasm are apoptosis-inducing factor (AIF), second mitochondria-derived caspase activator (Smac)/direct IAP-binding protein low pI (DIABLO), and Omi/high-temperature demand protein A (HtrA2). The consequence of the release of cytochrome c is the activation of caspase-3 through a complex known as the apoptosome. Smac/DIABLO or Omi/HtrA2 promote the activation of caspases by connection to the inhibitor of apoptosis proteins (IAP), which consequently leads to the disruption of its interaction with caspase-9 or -3 [16,19].

The third pathway of apoptosis activation is the one related to endoplasmic reticulum stress. The endoplasmic reticulum (ER) is the site of protein folding, lipid and sterol synthesis, and free calcium storage. Under stress in the ER, the level of mutant proteins increases, leading to a disruption in the equilibrium between the endoplasmic reticulum’s protein folding capacity and the demand for properly folded proteins. This phenomenon triggers ER stress, which is detected and managed by the unfolded protein response (UPR). Protein kinase RNA-like ER kinase (PERK), inositol-requiring protein 1 (IRE1a), and activating transcription factor 6 (ATF6) become activated once the concentration of misfolded proteins reaches a critical threshold. Apoptosis is induced here by the activation of Bax and Bak, two major pro-apoptotic proteins [19].

The unrestricted growth of cancer cells is caused, among others factors, by the ability to avoid apoptosis mechanisms, which include an imbalance of pro-apoptotic and anti-apoptotic processes, impaired caspase functions, and death receptor signaling [16]. Therefore, the factors involved in the regulation of apoptosis have a great diagnostic and interventional value in the treatment of diseases [15]. Numerous studies indicate the importance of flavonols as potential modulators of the apoptosis process in cancer cells (Table 2).

### 2.1. Quercetin

Studies conducted using rat models indicate the effect of QUE on colorectal cancer. The authors indicate an increase in the expression of genes for pro-apoptotic proteins, including caspase-3, and a decrease in the expression of anti-apoptotic genes, including Bcl-2, after the use of QUE, thus indicating its influence through the regulation of the internal mitochondrial pathway [71]. Later studies conducted on the SW48 colorectal cancer line also showed induction of apoptosis in cells after the use of QUE [72]. These reports are confirmed by studies using breast cancer cells (MCF-7) in the presence of the apoptosis inhibitor ZVAD. After adding QUE, an increase in the expression of Bax and caspase-3 and a decrease in the expression of the Bcl-2 gene were observed [20]. The effect on the expression of Bax and Bcl-2 was also demonstrated in HL-60 leukemia cells, and the authors of the study suggest that QUE induces apoptosis in the caspase-3-dependent pathway by inhibiting the expression of Cox-2 [21]. Apoptosis has also been found to be stimulated in gastric adenocarcinoma cells, as well as in non-small-cell lung cancer, and this apoptosis is inhibited by a p38 kinase inhibitor, a JNK inhibitor, and an ERK inhibitor [22,73]. The study performed on nine different cell lines confirmed previous reports, indicating the induction of apoptosis in the cancer cell lines CT-26 (colon cancer), PC-12 (pheochromocytoma), LNCaP (androgen-sensitive cancer line), and PC-3 (androgen-insensitive cancer line) [74]. Therefore, previous studies indicate the promotion of QUE-induced apoptosis via mitochondria [75]. QUE can also support conventional treatment methods. Studies performed using breast cancer cells showed that the addition of QUE to doxorubicin enhanced the induction of apoptosis in T47D cells [76]. In Figure 2, the impact of flavonols on the regulation of apoptosis pathways is depicted.

### 2.2. Kaempferol

Studies conducted on human cervical cancer cells suggest stimulation of apoptosis using KEM by reducing the expression of the anti-apoptotic genes PI3K, AKT, and Bcl-2, and a simultaneous increase in the expression of pro-apoptotic genes such as p53, p21, caspase3, caspase9, and Bax. As a result of these changes, the Bax/Bcl-2 ratio increases, which suggests stimulation of the intrinsic apoptosis pathway with an accompanying change in mitochondrial function [23]. Exposing head and neck cancer cells to KEM significantly led to the induction of apoptosis, accompanied by a decrease in Bcl-2, an increase in the concentration of caspase-3, and an increased release of cytochrome c [24]. These reports are confirmed by studies conducted on OVACAR-3 ovarian cancer cells. KEM increases the expression of not only caspase-3 and -9 but also -8. Moreover, the authors suggest the possible mediation of MEK/EK and STAT3 signaling in the induction of the process. KEM administration causes a concentration-dependent decrease in the phosphorylation of p-MEK and p-ERK and, therefore, a decrease in the expression of total MEK and ERK along with a decrease in phosphorylated pSTAT3 [25]. Studies on hepatocellular carcinoma cells enriched with molecular docking will confirm previous reports. The activity of KEM underlying its apoptosis-promoting effects may be related to its ability to regulate the expression levels of BAX, CDK1, and JUN proteins [27]. In colorectal cancer cells, KEM has additionally been found to increase the level of the cell membrane-bound FAS ligand, which increases sensitivity to the pro-apoptotic effect of anti-TRAIL antibodies in the human chronic myeloid leukemia cell line, thus sensitizing them to TRAIL, which may indicate the induction of apoptosis through the activation of death receptors on the cell surface [30,77]. Gao et al. [26] suggest that the induction of apoptosis with the participation of KEM occurs both through the extrinsic (via receptors) and intrinsic (via mitochondria) apoptotic pathways. According to the results of tests conducted on A2780/CP70 ovarian cancer cells, KEM significantly increased the cleavage of PARP-1 and the activity of caspase-3/7, as well as the activity of caspase-8 and -9 common to both pathways. Further analysis showed that KEM increases the expression of two death receptors, DR5 and Fas, and the adapter protein FADD [26].

Studies conducted on pancreatic cancer cells indicate another way of activating apoptosis after the use of KEM. It promoted apoptosis in cells by increasing the production of reactive oxygen species (ROS), which is involved in Akt/mTOR signaling [28]. In turn, after culturing colon cancer cells in the KEM environment, an increase in the level of ROS and, consequently, the induction of apoptosis was noticed. However, after the use of N-acetylcysteine as a ROS blocker, a weakening of caspase-3 and PARP cleavage and p38 phosphorylation was observed. The authors suggest that KEM induces apoptosis through ROS and p53-dependent p38 activation [32]. After the application of KEM to non-small-cell lung cancer cells associated with Nrf2 (a factor contributing to the development of treatment resistance), intracellular ROS accumulation was demonstrated, which resulted in the initiation of caspase-dependent apoptosis [33]. These studies indicate the importance of reactive oxygen species in the mechanisms of apoptosis activation with the participation of KEM.

The pro-apoptotic effect of KEM has also been demonstrated in prostate cancer, bladder cancer, and breast cancer cells [29,31,78,79].

### 2.3. Galangin

Galangin (GAL) is a flavonol that has pro-apoptotic properties. Studies performed on breast cancer cells have shown that GAL can effectively stimulate cells to induce TRAIL via the TRAIL/caspase-3/AMPK signaling pathway. The use of GAL significantly increases AMPK phosphorylation, cleavage of DR4, caspase-3, and -9, as well as the level of Bax protein, while decreasing the level of the anti-apoptotic protein Bcl-2 [34]. Moreover, according to research conducted on glioblastoma cells, the level of caspase-7 increases after the use of GAL [80]. The results obtained from an experiment performed on retinoblastoma cells indicate that GAL can increase the expression of PTEN and caspase-3 while reducing Akt phosphorylation, which results in the inhibition of proliferation and induction of apoptosis [39]. Similarly, in renal cancer cells, GAL increases the expression of Bax and Cyt-c and decreases the expression of Bcl-2 [36]. GAL can also suppress the expression of some important proteins of the PI3K/AKT signaling pathway [35,36,37]. Another possible pathway for the initiation of apoptosis by the compound is the JAK2/STAT3 pathway. Studies performed on gastric cancer cells indicate the effect of reversing the abnormal expression of proteins such as p-JAK2, p-STAT3, Bcl-2, cleaved caspase-3, cleaved PARP, and Ki67 by GAL. Moreover, GAL increases the accumulation of ROS and decreases Nrf2 and NQO-1, but increases HO-1, and this accumulation can be inhibited by STAT3 overexpression [38]. The accumulation of ROS accompanying apoptosis was also noticed after the application of GAL on kidney cancer cells in a dose-dependent manner [40]. The apoptosis process may also be mediated by the p53 protein. Studies performed on hepatocellular carcinoma cells after the use of GAL indicate reduced expression of miR675 and H19, leading to the initiation of apoptosis by stimulating the expression of the p53 protein [81]. Application of GAL to ovarian cancer cells also showed stimulation of the p53-dependent extrinsic apoptosis pathway through upregulation of the DR5 protein. This confirms the significant role of p53 in the induction of compound-induced apoptosis [82].

It is possible to use GAL in the treatment of cisplatin-resistant lung cancer. The simultaneous use of GAL and cisplatin leads to the cleavage of caspase-8, caspase-9, caspase-3, and cytochrome c, resulting in the induction of apoptosis. Further analysis showed an increase in the number of p-STAT3-, p-NF-κB, and Bcl-2-positive cells after GAL treatment. The authors suggest that the use of GAL may enhance cisplatin-based therapies in treatment-resistant lung cancer cells by inactivating the p-STAT3/p65 and Bcl-2 pathways [83]. The simultaneous use of GAL and TRAIL in the treatment of kidney cancer also gives promising results. Studies show that the simultaneous use of these two compounds leads to significantly increased induction of apoptosis in cancer cells showing signs of resistance to treatment [84].

### 2.4. Myricetin

Myricetin (MYR) is another flavonol with pro-apoptotic effects. After applying the compound to breast, thyroid, and gastric cancer cells, an increased level of split PARP and Bax proteins and a decreased level of Bcl-2 protein were observed [41,42,43]. Furthermore, MYR triggers caspase-3/-8/-9 activation and the release of AIF from cells, implying its role in inducing cancer cell death, partly via the activation of a caspase-dependent pathway [43,44,46]. Additionally, the expression levels of phosphorylated c-Jun N-terminal kinase (p-JNK) and phosphorylated mitogen-activated protein kinases (p-p38) were found to be increased, accompanied by a decrease in p-ERK [41]. Additionally, dose-dependent reductions in p-PI3K, p-Akt, and p-mTOR were found, which may indicate the involvement of the compound in the inhibition of the PI3K/Akt/mTOR pathway during apoptosis [42,47,85].

After culturing lung cancer cells in the presence of MYR, increased ROS production was noticed. This study revealed increased expression of P53 with an accompanying decrease in EGFR in cells treated with MYR [48]. Similar results were obtained after the use of MYR on ovarian cancer cells (OVCAR-3), in which the expression of p53 and p21 proteins was increased, which may indicate its involvement in the induction of apoptosis [45].

### 2.5. Isorhamnetin

Studies conducted on many types of cancer prove the role of ISO in the regulatory processes of apoptosis. The basic parameters that change after the use of ISO are elevated expression of Bax and caspase-3, along with reduced expression of Bcl-2 [49,50,53].There is also an increase in the activity of caspase-8 and -9, as well as an increase in the release of cytochrome c from the mitochondrion to the cytosol [51,52,53,86]. Similarly to the previous flavonols, ISO can also regulate apoptosis through the PI3K/Akt pathway. Studies indicate a reduction in the expression of Akt and the phosphorylated proteins PI3K (p-P13K) and Akt (p-Akt) [49,54,55,56,87]. Moreover, among the reports, there is information about the connection between isorhamnetin-induced apoptosis and increased expression of the Fas ligand [74,75].

### 2.6. Fisetin

The results of studies on ovarian cancer and uveal melanoma cells treated with FIS indicate changes in the levels of anti-apoptotic proteins, such as BCL2 and BCL-x, and pro-apoptotic proteins, such as BID, BAD, BAK, and BAX [57,58].Under the influence of Gal, the level of cleaved forms of caspase-8, -9,and -3, cytochrome c, and apoptosis-inducing factor (AIF) is also increased [57,58,59,65,88,89]. Studies performed on pancreatic cancer cells indicate the PI3K/AKT signaling cascade as a possible candidate in the initiation of apoptosis [60,63,66]. Additionally, Sabarwal et al. found that FIS caused an increase in total p53 in gastric cancer cells and is activated by phosphorylation at the S15 position, indicating the likelihood of its involvement in DNA damage [61].Subsequent studies involving gastric cancer cells with FIS showed a reduction in ERK activation 1/2 in a concentration-dependent manner, suggesting the involvement of another pathway in the initiation of apoptosis [62].

Further studies showed an increase in the amount of ROS and stimulation of intracellular Ca^2+^ secretion after culturing cells in a FIS environment, indicating that ROS could represent an alternative pathway for initiating apoptosis in cancer cells [57,64,88].

### 2.7. Morin

Studies conducted on chronic myeloid leukemia (CML) K562 and KCL22 cell lines demonstrated significant anticancer effects of MOR by suppressing the PI3K/AKT signaling pathway. The results indicate a reduction in phosphorylated AKT levels due to elevated PTEN expression, consequently resulting in the suppression of AKT signaling [67]. In addition, after the use of the compound, there was an increase in the level of Bax protein and a decrease in Bcl2. These results are confirmed by studies using colorectal cancer and melanoma cells in which, after MOR treatment, the expression of the Fas receptor and the activation of caspase-8, -9, and -3 were increased [68,69,70]. MOR induced apoptosis, which was correlated with the increased level of creation of reactive oxygen species and loss of cell mitochondrial membrane potential. These results indicate the action of MOR through both the internal and external apoptosis pathways [69].

## 3. Autophagy

Autophagy is a kind of cell death, during which cellular contents are surrounded by autophagic vesicles which, after combining with lysosomes, participate in the degradation of the contents. Its main role is to maintain homeostasis by eliminating dysfunctional organelles and protein aggregates; under certain conditions, it can mediate cell death. Three types of autophagy have been identified, including macroautophagy, microautophagy, and chaperone-mediated autophagy (CMA) [90].

The two main modes of autophagy-related cell death are:(1)ADCD (autophagy-dependent cell death), which is independent of other forms of programmed death.(2)AMCD (autophagy-mediated cell death), where autophagic molecules interact directly with cell death molecules or where the autophagy is related to other cell death mechanisms such as apoptosis, necrosis, or ferroptosis through its dynamic processes.

Cell deaths associated with autophagy can be induced simultaneously and interpenetrate each other under certain conditions, together with apoptosis, necroptosis, and ferroptosis [91].

Microautophagy

Microautophagy is the least known type of autophagy, in which lytic organelles independently take up components to be degraded from the cytoplasm. This type of autophagy is involved in the regulation of biosynthesis, transport, metabolic adaptation, organelle remodeling, and control over the quality of cellular components [92].

Macroautophagy

Macroautophagy is the best-known type of autophagy, in which autophagosomes deliver cytoplasmic components to be degraded to endosomes or lysosomes. In the first stage of autophagy, a portion of the cytoplasm is enclosed by an isolating membrane called the phagophore. The Atg9 protein plays a crucial role in its formation by supplying the necessary lipid components. This process is regulated by the Atg1 and Atg9 proteins, and a complex containing phosphatidylinositol 3-kinase. In the subsequent stage, two conjugation processes occur. In the first one, Atg12 is activated with the involvement of the Atg7 protein. Then, the Atg12 protein is transferred to the Atg10 protein, leading to the covalent binding of Atg12 to Atg5. The Atg12–Atg5 complexes then associate with Atg16L protein. The resulting complex, ATG12–ATG5–ATG16L1, plays a crucial role in the formation of the autophagosome. On the other hand, the second conjugation process involves the Atg3, Atg4, and Atg7 proteins, as well as the LC3 protein. The proLC3 form is proteolytically cleaved by the Atg4 protease, resulting in the formation of the LC3-I form. Then, the Atg7, Atg3, and Atg12–Atg5–Atg16L proteins are attached. In the next step, the LC3-I protein can bind with the highly lipophilic phosphatidylethanolamine (PE) and form the LC3-II form. These processes lead to the formation of the autophagosome, enclosing a portion of the cytoplasm along with proteins inside it. In the subsequent stage, the outer membrane of the autophagosome fuses with the lysosome, resulting in the formation of an autophagolysosome. Here, enzymatic digestion of the inner membrane of the autophagolysosome and its contents occurs under the influence of lysosomal enzymes [90,93].

Chaperone-mediated autophagy (CMA)

CMA enables the removal of specific damaged proteins when subjected to conditions of long-term starvation or oxidative stress. The CMA mechanism involves the formation of a bond between the chaperone complex and the target motif in the protein, which is then transferred to the lysosomes. After the complex reaches the lysosome, it interacts with the cytoplasmic tail of lysosome-associated membrane protein type 2A (LAMP-2A). After binding, the complex is moved to the lysosomal matrix, where it is completely degraded [93].

It is also worth remembering that there is a network of connections between autophagy and apoptosis. Beclin is a protein that links apoptosis and autophagy by its ability to bind to anti-apoptotic and pro-apoptotic proteins. Anti-apoptotic proteins such as Bcl-2 and Bcl-XL act as autophagy inhibitors, while pro-apoptotic proteins inhibit the interaction between beclin 1 and Bcl-2, thereby inducing autophagy. It has been shown that caspases 3, 7, and 8, which play the main role in the apoptosis process, have the ability to proteolyze beclin 1, which prevents the induction of autophagy. It was also found that calpain cleavage of the Atg5 protein involved in the formation of autophagosomes activates apoptosis. Moreover, autophagy can inhibit apoptosis partly by degrading active caspase 8 and inhibiting the activation of the Bid protein by beclin 1. Interconnections between the processes of apoptosis and autophagy also occur through the p53 protein. This protein is an inhibitor of cancer transformation. It stimulates the apoptosis process, but can also influence the autophagy process by stimulating or inhibiting it depending on its location in the cell. The fraction located in the cytoplasm inhibits the autophagy process by inducing mTOR1, while the nuclear fraction participates in stimulating the process [90,94].

Previous findings have suggested a dual role for autophagy in cancer; we now know that autophagy inhibits tumor initiation, but evidence suggests that its processes in some cancers depend on the stage of the disease and oncogenic mutations. Sometimes, it supports uncontrolled cell growth and the resulting increased metabolic activity. Moreover, autophagy affects the regulation occurring in cancer cells (Table 3), contributing to their growth and creating drug resistance [95,96].

### 3.1. Quercetin

Research has shown that QUE triggers autophagy by deactivating the Akt-mTOR pathway. The simultaneous use of the autophagy inhibitor 3-Methyladenine and the Akt-mTOR pathway inducer IGF-1 has shown that QUE inhibits cell motility and glycolysis by triggering autophagy through the Akt-mTOR pathway [97,98]. The analysis of cellular autophagy regulatory proteins and neuroglioma cell suppressors after the use of QUE nanoparticles showed not only a decrease in the expression of activated PI3K/AKT mTOR and Bcl-2 but also an increase in LC3, ERK, and cytoplasmic p53. This supports the theory regarding the underlying molecular mechanisms of autophagy induction via suppression of AKT/mTOR signaling and activation of the LC3/ERK/caspase-3 pathway [100]. Studies performed on primary lymphoma cells showed, in addition to suppression of the PI3K/AKT/mTOR pathway, also suppression via STAT3, and consequently a reduction in the expression of cellular proteins promoting survival, such as c-FLIP, cMyc, and cyclin D1 [117]. Moreover, after QUE treatment, p-AMPK expression increases with a concomitant decrease in p-mTOR expression in a dose-dependent manner. The mechanism of this phenomenon is the phosphorylation of AMPK, which in turn is an inhibitor of mTOR phosphorylation [99].

In the autophagy process, the LC3II/I ratio serves as an indicator of autophagy levels, while the level of p62 protein exhibits an inverse correlation with it. Studies on the effect of QUE on hepatocellular carcinoma cells showed that LC3II/I increases with the concentration of the compound used, while p62 decreases. The authors suggest that the activation of the NF-κB pathway, through its impact on p62 levels, could stimulate autophagy, as indicated by the decreased levels of IκBα and phosphorylated p65 [118]. These results are confirmed by studies carried out on lung cancer cells, where the use of QUE reduces the expression of the p62 protein and increases GFP-LC3B in a dose-dependent manner [119]. Additionally, it increases the mRNA levels of the autophagy-related protein 5, 7, and 12, LC3-II, and beclin 1 proteins [101]. Figure 3 details the principal autophagy pathways and shows the modulation exerted by flavonols on their regulation.

### 3.2. Kaempferol

The use of KEM on non-small-cell lung cancer cells showed an inhibitory effect of the compound on PI3K, AKT, and mTOR in a dose- and time-dependent manner. This proves the involvement of KEM in the suppression of the PI3K/AKT/mTOR pathway, thus leading to the induction of autophagy [102]. This is confirmed by studies using liver cancer cells. After the use of KEM, increased levels of autophagy-related protein 5, 7, and 12, p-AMPK, LC3-II, and beclin 1 proteins were observed, as well as decreased levels of cyclin B, cyclin-dependent kinase 1, p-AKT, and p-mTOR proteins [103,104]. KEM increases the conversion of LC3-I to LC3-II and leads to a decrease in p62 expression in cancer cells, suggesting the activation of autophagy [103,105,106,107,120].

KEM may also lead to stimulation of the autophagy process by activating IRE1-JNK-CHOP signaling, thus indicating that it may be a response to ER stress (endoplasmic reticulum stress). In turn, blocking ER stress inhibits KEM-induced autophagy, thus promoting prolonged cell survival [105]. Moreover, studies conducted on ovarian cancer cells indicate an association of ER stress with increased intracellular Ca^2+^ levels without changes in ROS levels, suggesting that Ca^2+^ disruption is one of the mechanisms by which ECM leads to the induction of autophagy and apoptosis mediated by the ER [106].

### 3.3. Galangin

The results of studies on the mechanism of action of GAL on glioblastoma cells indicate the involvement of the AMPK/mTOR pathway in GAL-induced autophagy through dephosphorylation of mTOR [80]. These reports are confirmed by studies involving laryngeal cancer cells. Additionally, a significant increase in the expression of LC3I, LC3II, and Beclin 1 induced by GAL exposure has been demonstrated [108,109,110]. After treatment of liver cancer cells with GAL, it was shown that the induction of autophagy enhanced SIRT1-LC3 binding and reduced the acetylation of endogenous LC3. The results indicate that GAL promotes the appearance of autophagic vacuoles, increases the expression of LC3 II, Beclin1, and the ratio of LC3 II to LC3 I, and decreases the expression of p62 in a time-dependent manner. Together, these findings point to a novel mechanism by which GAL induces autophagy through deacetylation of endogenous LC3 via SIRT1 [109].Increased p53 protein expression was detected in the same cells. It should be noted that the mutated p53 protein can inhibit autophagy in some cancer cells. This study revealed that GAL-induced autophagy was inhibited by p53 inactivation and enhanced by its overexpression, suggesting that GAL promotes autophagy through a p53-related pathway [110].

### 3.4. Myricetin

Studies conducted on four types of colon and gastric cancer cell lines showed that MYR modulates cell apoptosis and autophagy by suppressing the PI3K/Akt/mTOR signaling pathway and increasing the LC3-II/β-actin ratio and Beclin-1/β-actin expression [42,47,111].

The dependence of ER stress and autophagy caused by the action of MYR in liver cancer cells was demonstrated. They showed a significantly increased LC3-II/LC3-I ratio and decreased p62 protein levels in a dose-dependent manner. Furthermore, it was shown that MYR-induced autophagic flux and IRE1α expression were significantly increased, as were Ca^2+^ levels. The authors therefore suggest two possible pathways for autophagy induction with the participation of MYR, the IRE1α-JNK and Ca^2+^-AMPK pathways [112]. Another possible pathway of MYR action involves inhibition of the p38 MAPK and Stat3 signaling pathways regulated by MARCH1. MYR reduces MARCH1 protein levels in Hep3B and HepG2 cells, and also reduces the level of MAPK and Stat3 [113].

### 3.5. Isorhamnetin

The results of studies on the mechanism of action of ISO on gastric cancer cells showed the effect of the compound on PI3K and the blockade of the PI3K-AKT-mTOR signaling pathway. The use of ISO increased the level of p62 in cells and significantly reduced the expression of p-PI3K, p-AKT, and p-mTOR [87].

### 3.6. Fisetin

Application of FIS to prostate cancer cells inhibits mTOR activity and reduces the levels of Raptor, Rictor, PRAS40, and GbL, resulting in the loss of formation of mTOR complexes (mTORC)1/2. FIS also activates the mTOR repressor TSC2 by inhibiting Akt and activating AMPK [114]. In turn, FIS treatment of oral squamous cell carcinoma cells with Ca9-22 showed an increase in markers such as Beclin-1 and ATG5, and a decrease in the p62 conversion of LC3-I into LC3-II in a dose-dependent manner [115].

Studies conducted on the effect of FIS on pancreatic cancer using mouse models showed an enhancement of the AMPK/mTOR pathway, with no effect on autophagy, after the addition of an AMPK inhibitor compound. The authors suggest the existence of another autophagy regulation pathway. Research demonstrated that the stress-induced transcription factor p8 was upregulated in FIS-treated PANC-1 cells, and silencing of p8 hindered FIS-induced autophagy. It was established that p8-dependent autophagy is independent of AMPK, and that p8 regulates ATF6, ATF4, and PERK in response to ER stress through p53/PKC-α-mediated signaling. Furthermore, blocking the AMPK/mTOR pathway using the compound might augment p8-dependent autophagy. A new model for the action of FIS is proposed that is mediated by p8 via the p53/PKC-α pathway, which in turn affects the levels of PERK, ATF4, and ATF6 [116].

### 3.7. Morin

Studies on the effect of MOR hydrate in combination with cisplatin on hepatocellular carcinoma cells showed a reduced level of LC3I/II expression and an increase in p62 expression. Additionally, the levels of associated autophagy markers, including PI3KIII, Atg5, Atg7, and BECN-1, exhibited a notable increase in HepG2 cells, followed by a decrease subsequent to treatment with the combination of MOR and cisplatin. Further evaluation of the expression levels of the main autophagy-inducing markers (BECN-1 and LC3) at the transcriptional level revealed reduced BECN-1 and LC3 mRNA expression in CP-treated HepG2 cells, which was significantly reduced in CP-MOR-treated cells. These results suggest that the combination of CP-MOR acts as a substantial regulator of autophagy and a potent inducer of apoptosis, thereby helping to maintain homeostatic balance in HepG cells [121].

## 4. Pyroptosis

Pyroptosis is a type of cell death described as caspase-1 dependent. It is associated with infection with Salmonella and Shigella bacteria. Initially, it was considered part of the apoptosis pathway, but so far, a number of differences between the morphological features of pyroptosis and apoptosis have been demonstrated (Figure 1). Apoptosis is a programmed cell death without an ongoing inflammatory process, while pyroptosis may cause inflammation activated by extracellular or intracellular stimulation, which includes drugs, bacterial and viral infections, toxins, or chemotherapy. In cells undergoing pyroptosis, chromatin condenses and DNA fragments, but their nuclei remain intact. DNA damage relies on the activation of caspase-activated DNase (CAD) and the inhibition of inhibitor of CAD (ICAD) in apoptotic cells. However, CAD is dispensable during pyroptosis, despite the potential cleavage by caspase-1. Additionally, inflammation-induced pore formation leads to swelling and osmotic lysis in pyroptotic cells. The development of pores in the plasma membrane is contingent upon caspase-1 activation, leading to increased cell membrane permeability. The inflow of water from the pores causes cell swelling and osmotic lysis [122,123].

Pyroptosis is associated with the innate and adaptive immune systems and contains a variety of molecules. Members of the gasdemin family constitute the core of pyroptosis, and can be cleaved and activated by inflammatory caspases (caspase-1, -4, -5, -11), as well as those associated with apoptosis (caspase-3, -6, -8) and granzymes (granzyme A, granzyme B). Next, cytokines and alarmins are released from the formed pores, influencing the downstream pathway. Another important factor is the inflammasome. In addition to the above main components, there are also many other regulators operating in each part of the pathway [124].

Recent research indicates that pyroptosis may serve as a new cancer elimination strategy by inducing pyroptotic cell death and activating intense anti-tumor immunity [125].

### 4.1. Kaempferol

Studies conducted on glioblastoma multiforme cells showed an increased level of GSDME cleavage and an increase in the mRNA expression of the pro-inflammatory factors IL-1β and ASC 24h after the use of KEM. To elucidate the interplay between autophagy and pyroptosis, ROS generation and the autophagy pathway were blocked using the antioxidant reagent *N*-acetyl-l-cysteine (NAC) and the PI3K inhibitor 3-methyladenine (3-MA). NAC reduced the ROS level induced by KEM in cells. Furthermore, it was found that NAC reversed both autophagy by reducing LC3 cleavage levels and pyroptosis by reducing GSDME cleavage levels, indicating that KEM-induced ROS production contributes to autophagy and pyroptosis in cancer cells [126]. In Figure 4, an illustration depicting the primary pyroptosis pathways and showcasing the impact of flavonols on their regulation is presented.

### 4.2. Galangin

Studies conducted on glioblastoma cells showed an increased level of GSDME after the use of GAL. Furthermore, morphological features are consistent with pyroptosis, including the characteristic bubbles in the cell membrane and the swelling typical of this process. LDH release is also significantly increased in cells, indicating that GAL treatment disrupts cell membrane integrity. The authors suggest that inhibition of pyroptosis increases nuclear DNA damage in glioma cells, indicating a possible impact of pyroptosis on the degree of apoptosis during treatment [80].

### 4.3. Myricetin

The use of MYR demonstrates an inhibitory impact on lung cancer cells by activating pyroptosis. The analysis of the expression of proteins from the Gasdermin family showed that this phenomenon is attributed to the cleavage of GSDME by activation of caspase-3. Further analysis of mitochondria and the endoplasmic reticulum showed that MYR could cause ER stress and increase ROS levels. Subsequent to the inhibition of caspase-12, there was a notable decrease in the expression levels of cleaved caspase-3 and cleaved GSDME. These results demonstrate that MYR induces lung cancer cell death mainly through pyroptosis induced by the ER stress pathway [127].

### 4.4. Fisetin

There is a suggestion that FIS serves a protective function and impacts the advancement of hepatocellular carcinoma. FIS-treated cells were assessed for caspase 1 activity and IL-1b expression. From the obtained results, it can be concluded that caspase 1 activity and IL-1b secretion were significantly decreased in a dose-dependent manner in cells after its application, suggesting that FIS inhibits the activation of the NLRP3 inflammasome. During observation, it was shown that under the influence of LPS, the cells swelled and were enlarged with many vesicular projections, and that FIS could alleviate the occurrence of pyroptosis to some extent [128].

## 5. Ferroptosis

Ferroptosis is characterized by two primary biochemical traits, the buildup of iron and lipid peroxidation, both reliant on reactive oxygen species for cell demise. Ferroptosis differs biochemically, morphologically, and genetically from other cell deaths, but most publications describe it as closest to necrosis. The characteristic features of the process are loss of cell membrane integrity, moderate chromatin condensation, and cytoplasmic organelle swelling. In certain instances, ferroptosis can occur simultaneously with cell rounding and an elevated presence of autophagosomes [129,130].

Classic activators of ferroptosis include erastin and RSL3, which inhibit the antioxidant system by increasing intracellular iron accumulation. In particular, transferrin promotes ferroptosis by mediating iron uptake via the transferrin receptor. Degradation of intracellular iron storage proteins or the solute carrier of the iron export transporter family member 1 (SLC40A1) increases iron accumulation through autophagy, initiating or enhancing ferroptosis [131,132]. In turn, iron leads to the generation of excessive amounts of reactive oxygen species (ROS) via the Fenton reaction, thereby increasing oxidative damage. Additionally, iron can enhance the activity of lipoxygenase (ALOX) or EGLN prolyl hydroxylase enzymes, which play crucial roles in lipid peroxidation and oxygen homeostasis. The interaction between systemic and local cellular iron regulation influences ferroptosis sensitivity. Targeting genes associated with iron overload or using iron-chelating agents is effective in inhibiting ferroptotic cell death. The reason why only iron, rather than metals that also generate reactive oxygen species (ROS) in the Fenton reaction, can induce ferroptosis remains unclear. One hypothesis suggests that iron overload activates particular downstream effectors that facilitate the induction of ferroptosis upon the production of lipid ROS [129].

Functionally, ferroptosis may be a necessary process for maintaining homeostasis or may be the cause of diseases and pathological conditions. It can be caused by various physiological and pathological conditions related to the generation of stress. Ferroptosis is an adaptive feature used to eliminate cancer cells. It plays a key role in inhibiting carcinogenesis by removing cells that are deficient in key nutrients in the environment or that are damaged by infection or stress [133,134].

### 5.1. Quercetin

Studies conducted on gastric cancer cells showed a QUE-induced decrease in cell activity, which was reinstated by the addition of the ferroptosis inhibitor Fer-1. This suggests that this compound inhibits cell activity by facilitating ferroptosis. Moreover, an increase in iron content in cancer cells was found after the addition of QUE, which was also reversed by Fer-1. Assessment of the expression of ferroptosis-related proteins showed that QUE suppressed the expression of xCT and GPX4. These findings indicate that QUE induces ferroptosis in gastric cancer cells by downregulating the expression of xCT-system molecules and elevating iron levels. QUE has also been observed to reduce the expression of xCT and GPX4 [135,136]. This implies the engagement of the NRF2/xCT pathway in promoting ferroptosis through QUE-SLC1A5 [135]. These reports are confirmed by the results of studies conducted on HEC-1-A endometrial cancer cells, in which QUE increased the intracellular level of reactive oxygen species in cells and also decreased the expression of glutathione peroxidase 4 (GPX4) [137].

Time-dependent reductions in FTL (ferritin light chain) and FTH (ferritin heavy chain) protein levels were also found following QUE treatment in various cancer types. This effect can be blocked by Baf-A1, a lysosomal inhibitor that enhances the essential role of the lysosome. According to these results, free iron content increased after QUE treatment, which was also prevented by lysosome inhibitors (Baf-A1) and peroxidation inhibitors (ferrostatin-1). At the same time, TFEB knockdown could effectively abolish QUE-induced intracellular lipid peroxidation, suggesting that QUE-induced lipid peroxidation was dependent on TFEB-mediated ferritin degradation by lysosomes and iron accumulation [136]. Quercetin promotes TFEB expression and nuclear transcription, induces the onset of iron death, and thus exerts a pharmacological effect on the killing of breast cancer cells [138].

### 5.2. Kaempferol

Studies conducted on hepatocellular carcinoma cells showed mutual regulation of KEM-induced ferroptosis and autophagy. The authors demonstrated that inhibitors of ferroptosis notably triggered AMPK phosphorylation in HepG2 cells. Moreover, ACC, which constitutes the downstream pathway of AMPK and LKB1, is phosphorylated by ferroptosis inhibitors. Furthermore, ferroptosis inhibitors enhance Beclin-1 expression and induce the conversion of LC3BI to LC3BII [139].

### 5.3. Galangin

After applying GAL to fibrosarcoma cells, it was found that it can be used as a ferroptosis inhibitor, which increases the viability of HT1080 cells with an RSL3 inhibitor, reduces the level of ROS and MDA lipids, and increases the expression of PTGS2 and glutathione peroxidase 4 (GPX4) mRNA. GAL treatment influenced the phosphorylations of the AKT, PI3K, and CREB proteins, and the ferroptosis-inhibiting effect of GAL was counteracted by the PI3K inhibitor LY294002. These findings indicate that GAL may exert antiferroptosis effects through activation of the PI3K/AKT/CREB signaling pathway [140].

## 6. Necroptosis

Necroptosis is a form of programmed cell death considered to be very similar to apoptosis and is associated with the involvement of death domain receptors (DR), including FAS and tumor necrosis factor receptor 1 (TNFR1), or PRRs (pathogen recognition receptors), which recognize adverse signals from the intracellular and extracellular microenvironments. Extracellular factors include chemical and mechanical trauma, inflammation, or infections. In contrast to apoptotic cell death, where dead cells are efficiently eliminated without compromising the integrity of the cell membrane, necroptotic cell death results in membrane rupture, leading to the release of intracellular contents [131,141].

The current best-understood pathway for the initiation of necroptosis is related to the TNF-α receptor. Further cell death signaling can occur in two ways. TNF-α can trigger the formation of complex I (a survival complex that signals through NF-kB). Nevertheless, when RIPK1 undergoes deubiquitination, the complex transitions into apoptotic complex IIa. In the absence of caspase-8 and with elevated RIPK3 levels, the complex transforms into IIb (also known as the necrosome). This necrosome contains death domain-related proteins RIPK1, RIPK3, and Fas, which enable the cell to undergo necroptosis by direct phosphorylation of the kinase domain-like protein (MLKL). Phosphorylation of MLKL causes a pore to form an oligomer that punctures the plasma membrane and causes subsequent cell death. Other downstream effectors of RIPK3 include mitochondrial serine/threonine protein phosphatase11 and Calmodulin-dependent protein kinase [142,143].

Studies indicate that necroptosis, in addition to its key role in infection and virus development, plays a role in the regulation of tumor biology, including oncogenesis, tumor metastasis, and tumor resistance to treatment [144]. In a study utilizing 60 cell lines, reduced levels of RIPK3 were observed in two-thirds of the samples, indicating that cancer cells can avoid necroptosis. Moreover, necroptosis is closely linked to the prognosis of numerous cancers. The Cox proportional hazards model showed that RIPK3 expression is an independent prognostic factor in colorectal cancer patients with respect to overall survival and disease-free survival. Recent studies have shown that the expression of RIPK1, RIPK3, and MLKL is associated with better overall survival in hepatocellular carcinoma [131]. Given the key role of necroptosis in tumor biology, necroptosis has emerged as a new target for anticancer therapy, and new compounds and many therapeutic agents have been thought to protect against cancer by inducing or modulating necroptosis [144].

### 6.1. Quercetin

Research conducted with cholangiocarcinoma cells demonstrated that co-administration of QUE with the Smac LCL-161 mimetic resulted in elevated pMLKL expression in cancer cells. Utilization of inhibitors targeting pivotal necroptotic proteins, such as necrostatin-1 (Nec-1), GSK’872 (GSK), and necrosulfamide (NSA), which specifically inhibit RIPK1, RIPK3, and MLKL, respectively, revealed that all inhibitors markedly attenuated cell death provoked by the combination treatment of QUE and the Smac mimetic LCL-161 in cells. These studies indicate the involvement of the RIPK1/RIPK3/MLKL necroptosis pathway after treatment in cholangiocarcinoma cells [145].

The involvement of necroptosis in QUE-induced cell death was confirmed in breast cancer cells. In the presence of the necroptosis inhibitor Nec-1, MCF-7 cell viability increased compared to the control. After QUE application, the expression of RIPK1 and RIPK3 genes increased significantly. The authors suggest the involvement of QUE through the RIPK1- and RIPK3-dependent necroptosis pathway [20]. In Figure 5, an illustration portraying the key pathways of ferroptosis and necroptosis, along with the impact of flavonols on their regulation, is depicted.

### 6.2. Kaempferol

As in QUE, the application of KEM with the Smac mimetic LCL-161 on cholangiocarcinoma cells led to a significant increase in pMLKL expression. The results following inhibitors targeting key necroptotic proteins also showed that all inhibitors significantly inhibited cell death induced by combined treatment of KEM with the Smac mimetic LCL-161 in cells. It is concluded that KEM, like QUE, influences the induction of necroptosis via the RIPK1/RIPK3/MLKL pathway [145].

### 6.3. Fisetin

A2780 cells cultured with FIS supplemented with z-VAD showed a significant increase in ZBP1, RIP3, and MLKL protein levels compared to control cells, while the level of HMGB1, an inflammation-related marker, showed no significant difference between groups. Study results indicate that FIS-induced necroptosis in ovarian cancer cell lines is mediated by the ZBP1/RIP3/MLKL pathway [146].

## 7. Cuproptosis

Cuproptosis is a type of cell death induced by Cu accumulation in the mitochondria, leading to the aggregation of lipoylated dihydrolipoamide S acetyltransferase (DLAT). This leads to proteotoxic stress and consequently cell death. Cuproptosis occurs mainly in energy-producing cells that use oxidative phosphorylation (OXPHOS) as the main metabolic pathway. Cuproptosis is characterized by aggregation of lipoylated DLAT mitochondrial enzymes and loss of the iron-sulfur cluster protein (Fe-S) [147].

The disruption of copper homeostasis fosters cancer development and leads to irreversible cell damage. A combination of cuproptosis-targeting molecules including drugs or natural agents combined with existing therapies may open new possibilities for cancer treatment [148]. Nonetheless, current copper-containing agents exhibit limited targeting specificity and may induce significant side effects in patients undergoing treatment. These limitations hinder the development and clinical implementation of cancer treatment strategies based on cuproptosis mechanisms [149]. The use of flavonoids may maximize targeted cancer treatment while limiting toxic side effects.

Previous research, including the study conducted on neuroblastoma cells, has shown that QUE can support conventional treatment methods through its neuroprotective effect. QUE reduced Cu-induced neurotoxicity in the SH-SY5Y cell line [150]. However, research remains inconclusive. Unlike QUE, GAL enhances the toxic effects of copper and exacerbates cell death, while also helping to stimulate the production of ROS. The authors therefore suggest caution when considering potent antioxidants for adjuvant therapy in copper-related neurodegeneration [151]. Therefore, further research is needed to explain the impact of individual flavonols on the course of copper-induced death.

## 8. Protective Effects of Flavonoids against Cancer: In Vivo Evidence

Experiments on animals play a crucial role in preclinical research as they provide an excellent model for evaluating the effectiveness and safety of potential therapeutic agents before transitioning to clinical trials.

### 8.1. Quercetin

To assess the impact of QUE on tumor growth, a xenograft model of gastric gland adenocarcinoma was constructed, demonstrating that the tumor volume in the quercetin-treated group was significantly smaller than in the control group. Additionally, following quercetin treatment, levels of ROS and MDA significantly increased, while GSH levels decreased compared to the control group. Results indicated that after quercetin treatment, levels of TfR1, GPX4, and SLC7A11 significantly decreased. The findings suggest that the anticancer action of quercetin is associated with autophagy-dependent ferroptosis [152]. Studies using a mouse model of prostate cancer have shown that tumor growth was significantly inhibited due to the synergistic action of QUE and paclitaxel. Additionally, the research demonstrated that the expression of cleaved caspase-3 increased in the Que+PTX group compared to monotherapy, indicating that combination therapy may effectively induce apoptosis in cancer cells. Expressions of CHOP and GRP78 also noticeably increased, suggesting that the combined treatment effectively induced ER stress and ROS production, leading to cancer cell death [153]. The experimental mouse model of liver cancer demonstrated that treatment with QUE significantly reduced tumor volume. Additionally, the expression of PCNA was effectively decreased by QE treatment, while levels of Bax increased. These results suggest that QUE inhibits tumor growth in vivo through apoptosis [154]. Other studies have shown that QUE significantly reduces tumor size in a mouse model of liver cancer heterotransplantation. Additionally, it has been demonstrated that the level of HK2 was significantly reduced in the QUE-treated group. Furthermore, phosphorylated Akt and mTOR were significantly decreased, confirming that QUE inhibits HCC progression by inhibiting glycolysis via HK2 in vivo [155]. It has also been demonstrated that QUE inhibits liver tumor growth in an in vivo model partially through the stimulation of autophagy. Levels of LC3A/B and p62 proteins in tumor tissues were quantitatively assessed by immunohistochemistry, showing that QCT treatment significantly increased the level of LC3A/B protein and decreased the level of p62 protein in tumors compared to the control. These results indicate that QCT can induce autophagy in liver tumors in vivo [156].

### 8.2. Kaempferol

Using a mouse experimental model of breast cancer, it was demonstrated that the expression of the tumor-promoting gene CCND1 was significantly decreased compared to the control group. An increase in the expression of the Bcl2 gene was also observed. Additionally, it was shown that the pro-tumorigenic gene CyPA and the pro-autophagic gene BECN1 were significantly decreased after KEM administration [157]. Studies conducted on mice with oral cavity cancer showed that tissue samples from mice treated with KEM exhibited a significant increase in apoptosis compared to the control group. Analysis of the expression of proteins related to apoptosis, autophagy, and MAPK revealed that the expression of p-JNK was particularly high. Overall, KEM increased the expression of apoptotic and autophagic marker proteins and decreased the expression of inhibitory proteins. Additionally, compared to the control, reduced expression of p-ERK and increased expression of p-JNK and p-p38 were observed. In particular, among the MAPK pathway, the expression of p-JNK was higher. Therefore, the KEM-treated group indicated that the JNK pathway of the MAPK pathway influences the apoptosis and autophagy of cancer cells [104]. The impact of KEM on non-small-cell lung cancer showed that tumor growth was significantly inhibited in increased mitochondrial mass in breast precancerous lesions. Interestingly, KEM promoted the expression of p-MFF, PINK1, Parkin, and LC3II, and reduced the expression of TOM20. Thus, KEM induces mitochondrial fission and mitophagy in vivo [158]. Studies investigating the role of 17β-estradiol and Triclosan, in combination with KEM using an experimental mouse model with xenotransplantation of MCF-7 breast cancer cells, showed that administration of 17β-estradiol and Triclosan in combination with KEM resulted in a significant reduction in tumor volume. The combination treatment showed a decreased number of nuclei with incorporated BrdU and reduced expression of PCNA and cyclin D1. The expression of P21 showed an opposite pattern to the expression of PCNA or cyclin D1. The combination therapy significantly increased the expression of Bax and decreased the expression of cathepsin D protein [159].

### 8.3. Galangin

The results of studies conducted on mice have shown that GAL inhibits tumor growth by suppressing the expression of H19 in vivo and may be involved in the apoptosis of cancer cells. Additionally, there is evidence suggesting that GAL may induce apoptosis in cells by regulating the expression of the p53 protein [81]. Studies using a retinoblastoma xenograft model have demonstrated that treatment with GAL significantly reduces tumor size. Furthermore, GAL lowers the expression level of KI-67 while enhancing the cleavage of PTEN and caspase-3, and the expression of p-Akt is decreased by GAL. Additionally, it has been shown that GAL decreases the expression of PIP2 and PIP3 in tumor tissue samples in vivo [39]. To confirm the inhibition of human laryngeal cancer growth by GAL, studies were conducted using a mouse xenograft model of TU212. Upon GAL administration, both the volume and mass of the tumor were inhibited. Additionally, in the GAL-treated group, a reduction in Ki-67 and an increase in TUNEL levels were observed. These results indicate that GAL may promote the suppression of the growth of xenotransplanted human laryngeal cancer cells in vivo. The studies showed that GAL prevents proliferation, invasion, and migration of human laryngeal cancer by suppressing PI3K/AKT and p38, resulting in caspase activation, NF-κB dephosphorylation, and mTOR inactivation with decreased Ki-67 expression and increased TUNEL levels [108].

### 8.4. Myricetin

The impact of MYR in vivo was investigated using a subcutaneous xenograft model of A2780 cells in nude mice. The study demonstrated a reduction in tumor volume after administration of MYR at a dose of 100 mg/kg. Additionally, the research confirmed that oral administration of MYR led to an increase in Bax levels, a decrease in Bcl-2 levels, and a corresponding elevation in the Bax/Bcl-2 ratio. It was demonstrated that MYR inhibited metastasis to the liver and lungs. Furthermore, the expression of Ki-67, MMP9, and EGFR in the myricetin-treated group was reduced compared to the control group [160]. The inhibitory effect of MYR on NCI-H446 and A549 cancer cells was also investigated in an in vivo model by injecting them subcutaneously into nude mice and administering MYR. The results showed that the tumor size in the MYR-treated group was significantly smaller than in the control group, indicating that myricetin may effectively inhibit lung cancer cells in vivo. The results also demonstrated that GSDME and cleaved caspase-3 were significantly higher in the MYR-treated group compared to the untreated group. In summary, MYR may inhibit tumor growth and induce cell pyroptosis in vivo [127].

### 8.5. Isorhamnetin

Expression of Ki-67, caspase-3, and PD-L1 was analyzed using immunohistochemistry in mice with breast cancer treated with ISO. It was demonstrated that ISO inhibited the EGFR-STAT3-PD-L1 signaling pathway and blocked tumor development, significantly increasing the survival of healthy cells. The cell membrane receptor EGFR was identified as a direct target of ISO [161]. It was also demonstrated that ISO induces apoptosis in Ishikawa cells by triggering the endogenous mitochondrial apoptotic pathway and the exogenous death receptor pathway, promoting the endoplasmic reticulum stress-related pathway. Additionally, ISO affected the expression of proteins associated with MMP2 and MMP9 and suppressed metastasis. It was found that ISO resulted in a reduction in the proliferation marker Ki-67, indicating that ISO has a beneficial effect on mitigating the malignancy of cancer cells. Additionally, a decrease in the expression of caspase-3, MMP2, and MMP-9 was demonstrated, along with an increase in the endoplasmic reticulum stress protein Chop [86]. An in vivo mouse model of gallbladder cancer cells was also developed to elucidate the anti-tumor effects of ISO. The size and mass of the tumor significantly decreased after ISO administration. Furthermore, no visible loss of body weight or mortality was observed in mice undergoing ISO treatment, suggesting that ISO is safe for in vivo use and does not induce side effects. The results indicate that ISO treatment significantly increased the expression of cleaved PARP, p53, cleaved caspase 9, BAX, cleaved caspase 3, and p27, while reducing the expression of BCL-2, N-cadherin, Slug, CDK1, p-PI3KP85α/γ/β, and p-AKT1 in the tumor tissues. A decreased level of proliferation marker Ki-67 and p-AKT1 expression was demonstrated after isorhamnetin treatment. These findings suggest that isorhamnetin may inhibit GBC growth in vivo and induce apoptosis through the PI3K/AKT signaling cascade [87]. The results also indicate that ISO may downregulate Hsp70 genes and promote apoptosis in colon cancer cells [162]. The anticancer effect of ISO was also investigated in vivo on murine non-small-cell lung cancer cells (A549). In mice treated with ISO for two weeks, no adverse effects were observed, including no loss of body weight, mortality, or lethargy. The tumor size was significantly smaller in most mice treated with ISO at a dose of 0.5 mg/kg compared to the control group. It is worth noting that the tumor size was significantly smaller in the group that received simultaneous injection of 3-methyladenine (22.4 mg/kg) or chloroquine (10 mg/kg) compared to mice injected only with ISO. It was demonstrated that the apoptosis index significantly increased in groups receiving either ISO alone or ISO in combination with 3-methyladenine or chloroquine. Additionally, levels of cleaved caspase-3 showed a similar trend. Significant reduction in the proliferation index was also observed after ISO administration. These results confirmed that apoptosis through caspase activation is a key factor contributing to tumor growth inhibition and that suppressing autophagy significantly enhances the inhibitory effect of ISO on NSCLC [55]. It has also been shown that ISO inhibits viability, enhances the apoptotic effect of capecitabine, abolishes NF-κB activation, and suppresses the expression of various NF-κB-regulated gene products in cancer cells. In the xenograft model of gastric cancer transplantation, administration of ISO alone (1 mg/kg body weight, i.p.) significantly inhibited tumor growth, as well as in combination with capecitabine. ISO additionally reduces NF-κB activation and the expression of various proliferative and oncogenic biomarkers in tumor tissues [163]. It has also been demonstrated that after ISO treatment, the lung tumor mass in mice was significantly smaller, with tumor size being smaller than in the control group. Morphological studies of the tumors revealed typical condensed chromatin, cell shrinkage with condensed cytoplasms, and nuclear fragmentation characteristic of apoptotic cell death [164].

### 8.6. Fisetin

Research conducted on female C57BL/6 mice with subcutaneous lymphoma indicates that FIS inhibits tumor growth through the induction of cell apoptosis, inhibition of proliferation, and angiogenesis. Biochemical studies have shown no significant differences in liver and kidney function markers in female mice treated with FIS [165]. A preclinical study conducted on female Albino Swiss CD1 mice with Ehrlich tumors as a model of breast cancer showed that FIS treatment slightly inhibits tumor growth in mice. However, treatment with modified cholefitosomes containing FIS significantly inhibited the rate of tumor growth. Studies have shown that modified cholefitosomes had comparable cytotoxicity, significantly surpassing the cytotoxicity of free FIS. It has also been demonstrated that TGF-β1 and its associated non-canonical signaling pathways, ERK1/2, NF-κB, and MMP-9, were involved in halting tumor proliferation. Additionally, the investigated compounds exhibited a pronounced increase in the amount of E-cadherin compared to free FIS [166]. Other studies based on the same experimental model have also demonstrated the anti-tumor efficacy of FIS. Mice serving as positive controls showed a significant increase in tumor size throughout the study period. The inhibition of tumor growth was achieved following treatment with FIS, FIS-loaded nanosponges, and actoferrin-coated FIS-loaded β-cyclodextrin nanosponges. Furthermore, it was demonstrated that the positive control exhibited significant overexpression of CD1 levels compared to the negative control, whereas all treated groups showed a significant reduction in CD1 levels. Additionally, it was shown that FIS significantly lowered the level of Bcl-2 and increased the expression of the Bax gene. Moreover, loading FS into NS (nanosponges) resulted in a significant decrease in Bcl-2 levels along with increased expression of the Bax gene compared to FIS alone. All therapies significantly increased the expression of the caspase-3 gene, confirming the involvement of the apoptotic process in the anti-tumor activity of FIS [167]. To determine the effectiveness of the in vivo combination therapy of FIS and gemcitabine, researchers utilized mice with an orthotopic pancreatic cancer allografts. It was demonstrated that tumor sizes were significantly reduced in mice treated with FIS or gemcitabine individually, with the combination regimen showing the best effect. It has been demonstrated that the levels of CDK1, p-STAT3, CD44, and Sox2 decreased after treatment with FIS alone or in combination therapy, indicating that FIS-mediated inhibition of the CDK1–STAT3 axis contributed to increased chemosensitivity in vivo [168]. The impact of Zinc-FIS hybrid nanoparticles (ZFH) was evaluated using a model of oral cavity cancer induced by 4-nitroquinoline 1-oxide (4-NQO) in in vivo studies on rats. It was demonstrated that ZFH significantly reduced the levels of biomarkers specific to oral squamous cell carcinoma in serum, decreased the histological grade of the tumor, and increased the level of caspase-3.Incorporating FS into networked zinc nanoparticles significantly enhanced its ability to induce apoptosis in cancer cells [169]. Furthermore, to assess the impact of FIS on pancreatic tumor growth in vivo, a xenograft experiment was conducted on nude mice. It was demonstrated that the mean tumor volume along with mass significantly differed between the treatment groups and the control. Additionally, the expression of PI3K, p-AKT, and p-mTOR proteins was markedly reduced in the FIS-treated group. These data indicate that FIS inhibits pancreatic tumor growth in vivo [63]. The aim of the study was to delineate the in vivo mechanisms of oral administration of FIS, with a particular focus on mitochondrial dysfunction in lung tissues using benzo(a)pyrene as a model carcinogen for the lungs. Treatment with FIS led to a significant increase in the expression of Bax and a decrease in the expression of Bcl-2, along with a significant increase in the expression of caspase-3, caspase-9, and cytochrome c in mice with lung cancer [170]. The effectiveness of intravesical instillations of FIS was also investigated, and the fundamental mechanisms of FIS’s inhibitory action on bladder cancer were determined using an orthotopic rat model of bladder cancer induced by N-methyl-N-nitrosourea. It has been demonstrated that FIS-induced apoptosis in bladder cancer occurs through the modulation of two related pathways: upregulation of the p53 pathway and downregulation of NF-κB pathway activity, leading to alterations in the ratio of pro- and anti-apoptotic proteins. Meanwhile, administration of FIS significantly reduced the incidence of MNU-induced bladder tumors by suppressing NF-κB activation and modulating the expression of NF-κB target genes, which regulate cell proliferation and apoptosis. FIS treatment significantly decreased the expression of PCNA, Bcl-2, and cyclin D1, while increasing the expression of p21, p53, and Bax [171]. Figure 6 depicts a diagram delineating the influence of flavonols on cell death mechanisms across various cancer types, as evidenced by in vivo studies. The illustration highlights the regulatory role of flavonols in modulating these pathways, shedding light on their potential therapeutic implications in cancer treatment.

## 9. Conclusions

Cells originating from different types of tumors may exhibit sensitivity to different types of cell death, either through increased sensitivity or resistance to a particular mechanism. A thorough understanding of the sensitivity of individual tumor types to the action of a specific mechanism of cell death can play a crucial role in disease treatment by targeting regulations involving that particular type of cell death.

The information presented in the study demonstrates that apoptosis is the most well-understood mechanism of cell death through which flavonols act. It is worth noting that for some compounds such as kaempferol, the action may occur through a series of mechanisms involving nearly all described types of cell death. To harness the potential of compounds in treating a particular type of cancer, it is essential to thoroughly understand the resistance mechanisms characterizing that type of cancer and to apply the compound in a specific mechanism, aided by the use of inhibitors. Furthermore, the obtained research indicates the significance of the administered doses, which at certain concentrations can enhance the treatment with chemotherapeutic agents but become toxic to cells if the concentration is too high. It is therefore crucial to appropriately select the appropriate dose of flavonol for a particular type of cancer, taking into account the action of additional drugs and the resistance of the cells.

In summary, the aim of the study was to characterize various mechanisms of cell death and to present the potential regulation of certain signaling pathways by selected flavonols.

Despite numerous studies, the anticancer role and the molecular mechanisms underlying the regulation of apoptosis, autophagy, necroptosis, pyroptosis, ferroptosis, and cuproptosis mediated by flavonols remain incompletely elucidated and the results of studies conducted to date are inconclusive. It is worth noting that evaluating the mechanisms underlying the anticancer activity of these compounds in therapy targeting pathways of various types of cell death may prove useful in developing new therapeutic schemes and overcoming resistance to currently used treatments.

## Figures and Tables

**Figure 1 nutrients-16-01201-f001:**
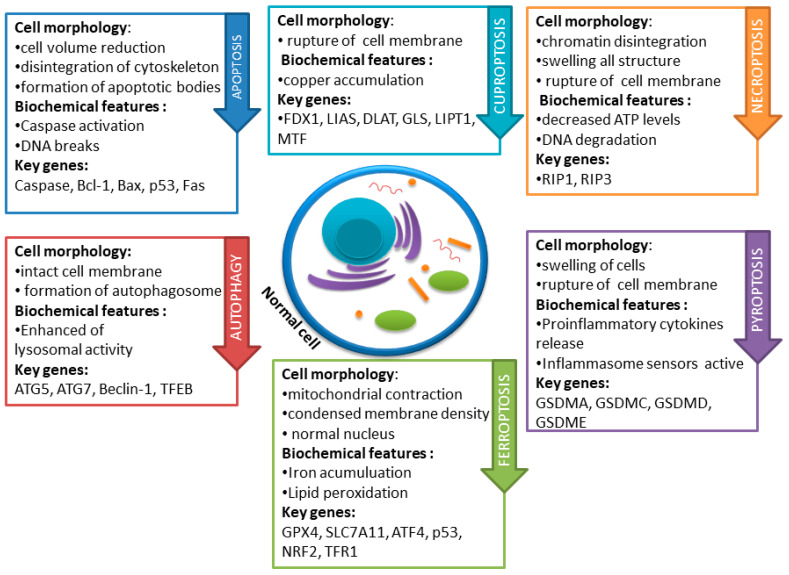
Basic differences in cell death mechanisms including apoptosis, autophagy, necroptosis, pyroptosis, and ferroptosis. Each pathway exhibits distinct molecular and morphological features, contributing to the intricate regulation of cell fate.

**Figure 2 nutrients-16-01201-f002:**
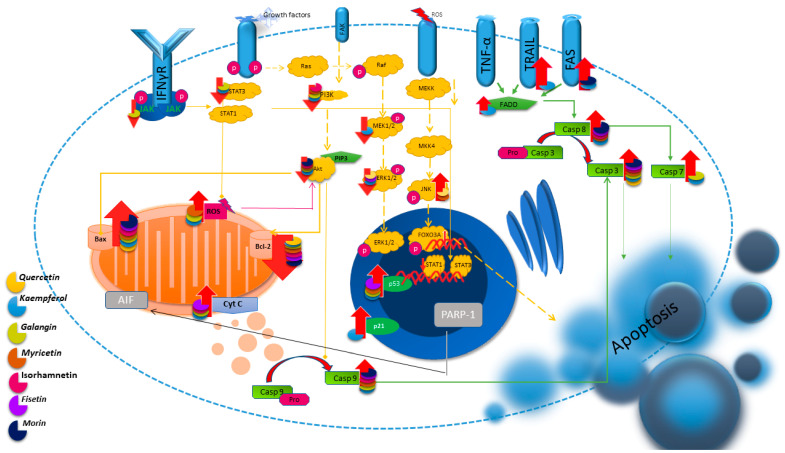
Illustration of the main apoptosis pathways and the regulatory influence of flavonols on them.

**Figure 3 nutrients-16-01201-f003:**
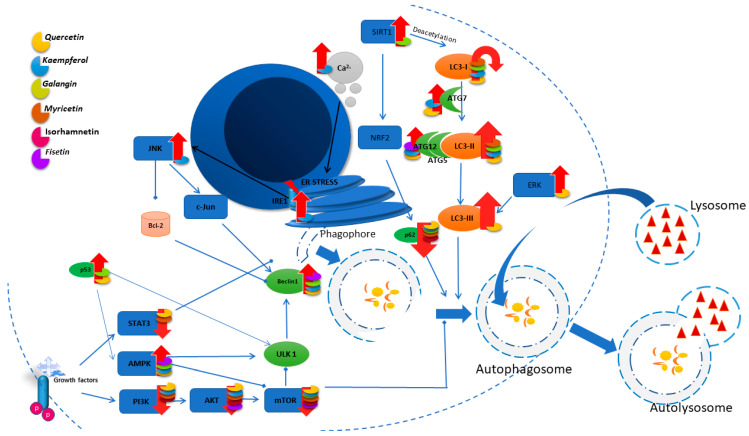
Illustration of the main autophagy pathways and the influence of flavonols on their regulation.

**Figure 4 nutrients-16-01201-f004:**
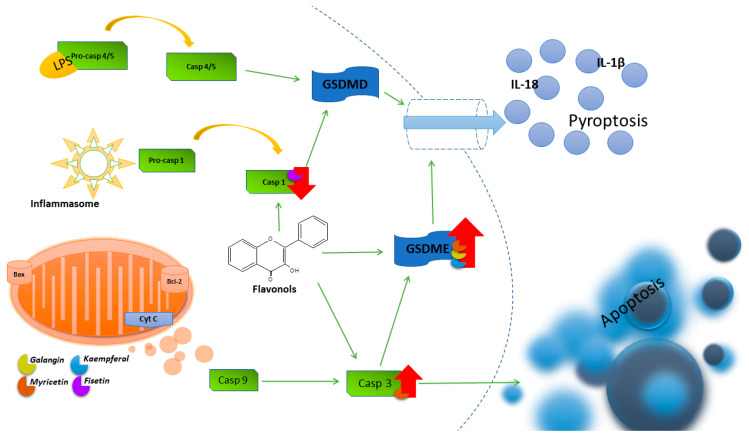
Illustration of the main pyroptosis pathways and the influence of flavonols on their regulation.

**Figure 5 nutrients-16-01201-f005:**
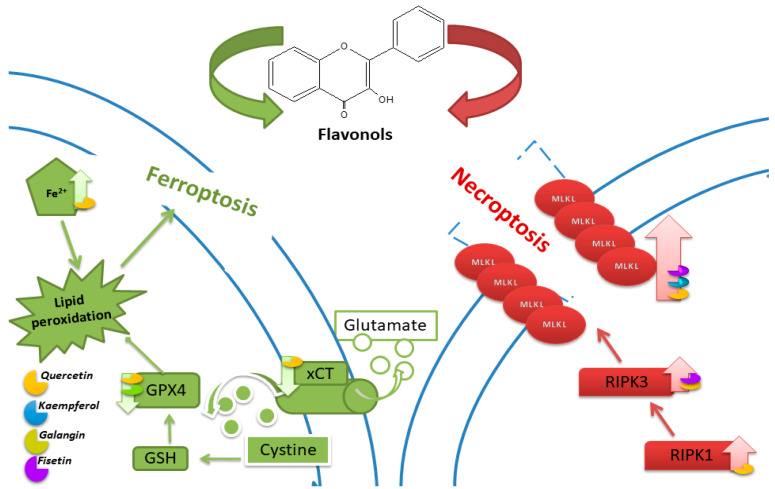
Illustration of the main ferroptosis and necroptosis pathways and the influence of flavonols on their regulation.

**Figure 6 nutrients-16-01201-f006:**
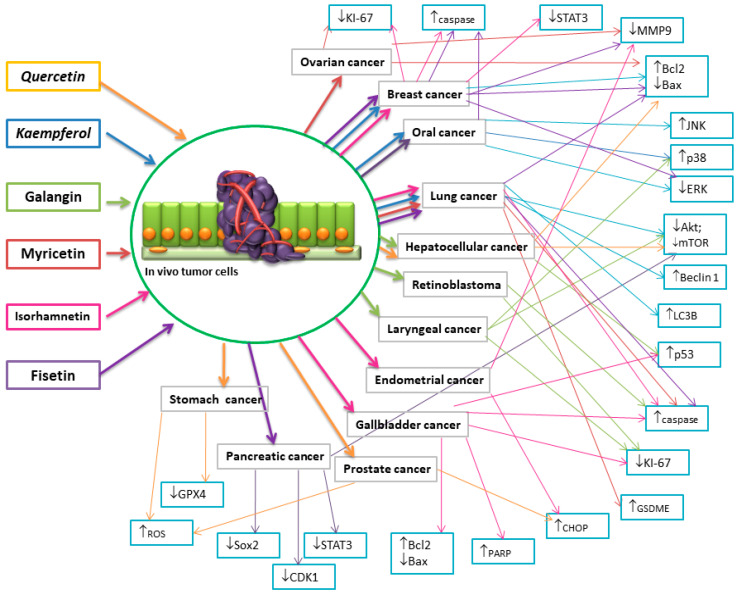
Diagram illustrating the effect of flavonols on cell death mechanisms in various types of cancers in vivo studies. (↑-increase, ↓ decrease).

**Table 2 nutrients-16-01201-t002:** Summary of the most important apoptosis induction pathways involving flavonols and their impact on individual types of cancer (↑-increase, ↓ decrease).

Flavonoid	Mechanism	Cancer Type	Cell Line	Ref.
Quercetin	↓ Bcl2 ↑ Bax	Breast cancer	MCF-7	[20]
		Leukemia	HL-60	[21]
	↑ caspases activation	Breast cancer	MCF-7	[20]
		Leukemia	HL-60	[21]
		Gastric cancer	AGS	[22]
	↑ JNK	Gastric cancer	AGS	[22]
Kaempferol	↓ Bcl2 ↑ Bax	Cervical cancer	HeLa	[23]
		Submandibular gland cancer	A253	[24]
		Ovarian cancer	OVACAR-3	[25,26]
		Liver cancer	HepG2	[27]
		Pancreatic cancer	PaCa-2, PANC-1	[28]
		Bladder cancer	5637, T24	[29]
	↑ caspases activation	Cervical cancer	HeLa	[23]
		Submandibular gland cancer	A253	[24]
		Ovarian cancer	OVACAR-3, A2780/CP70	[25,26]
		Colorectal cancer	HT-29	[30]
		Pancreatic cancer	PaCa-2, PANC-1	[28]
		Breast cancer	MDA-MB-231	[31]
	↓ PI3K, AKT	Cervical cancer	HeLa	[23]
	↑ p53	Cervical cancer	HeLa	[23]
		Colorectal cancer	HCT15, HCT116	[32]
		Bladder cancer	5637, T24	[29]
	↑ p21	Cervical cancer	HeLa	[23]
		Bladder cancer	5637, T24	[29]
	↑ Cytochrome c	Oral cancer	SCC-9	[24]
		Submandibular gland cancer	A-253	[24]
		Ovarian cancer	OVACAR-3	[25]
	↓ STAT3	Ovarian cancer	OVACAR-3	[25]
	↓ MEK, ERK	Ovarian cancer	OVACAR-3	[25]
	↑ FAS	Colorectal cancer	HT-29	[30]
		Ovarian cancer	A2780/CP70	[26]
	↑ ROS	Pancreatic cancer	PaCa-2, PANC-1	[28]
		Colorectal cancer	HCT15, HCT116	[32]
		Non-small-cell lung cancer	NSCLC	[33]
Galangin	↓ Bcl2 ↑ Bax	Breast cancer	MCF-7, T47D	[34,35]
		Kidney cancer	A498	[36]
		Nasopharyngeal cancer	NPC-TW039, NPC-TW076	[37]
		Gastric cancer	MGC 803	[38]
	↑ caspases activation	Breast cancer	MCF-7, T47D	[34,35]
		Retinoblastoma	Y-79, HXO-Rb44	[39]
	↑ TRAIL	Breast cancer	MCF-7, T47D	[34,35]
	↓ PI3K, Akt	Retinoblastoma	Y-79, HXO-Rb44	[39]
		Kidney cancer	A498	[36]
		Nasopharyngeal cancer	NPC-TW039, NPC-TW076	[37]
		Breast cancer	MCF-7	[35]
	↑ Cytochrome c	Kidney cancer	A498	[36]
	↓ STAT3	Gastric cancer	MGC 803	[38]
	↑ ROS	Gastric cancer	MGC 803	[38]
		Kidney cancer	Caki1, 7860	[40]
Myricetin	↓ Bcl2 ↑ Bax	Breast cancer	SKBR3,T47-D	[41]
		Gastric cancer	AGS	[42]
		Thyroid cancer	SNU-790 H, SNU-80 HATC	[43,44]
		Ovarian cancer	A2780/CP70 OVCAR-3	[45]
	↑ caspases activation	Thyroid cancer	SNU-790 H, SNU-80 HATC	[43,44]
		Breast cancer	T47-D	[46]
		Ovarian cancer	A2780/CP70 OVCAR-3	[45]
	↓ PI3K, Akt	Gastric cancer	AGS	[42]
		Colorectal cancer	HCT116, SW620	[47]
	↑ ROS	Lung cance	A549	[48]
Isorhamnetin	↓ Bcl2 ↑ Bax	Melanoma	B16F10	[49]
		Liver cancer	Hep3B	[50]
		Gastric cancer	AGS-1, HGC-27	[51]
		Breast cancer	MDA-MB-231, MCF-7	[52]
	↑ caspases activation	Melanoma	B16F10	[49]
		Bladder cancer	T24, 5637	[53]
		Liver cancer	Hep3B	[50]
		Gastric cancer	AGS-1, HGC-27, MKN-45	[51,54]
		Breast cancer	MDA-MB-231, MCF-7	[52]
		Non-small lung cancer	A549	[55]
	↓ PI3K, Akt	Melanoma	B16F10	[49]
		Prostate cancer	DU145, PC3	[56]
		Gastric cancer	MKN-45	[54]
	↑ Cytochrome c	Bladder cancer	T24, 5637	[53]
		Breast cancer	MDA-MB-231, MCF-7	[52]
		Gastric cancer	MKN-45	[54]
Fisetin	↓ Bcl2 ↑ Bax	Oral cancer	HSC3	[57]
		Melanoma	M17, SP6.5	[58]
		Non-small lung cancer	NCI-H460	[59]
		Breast cancer	4T1	[60]
		Gastric cancer	AGS, SNU-1, SGC790	[61,62]
	↑ caspases activation	Oral cancer	HSC3	[57]
		Melanoma	M17, SP6.5	[58]
		Non-small lung cancer	NCI-H460	[59]
		Pancreatic cancer	PANC-1	[63]
		Gastric cancer	AGS, SNU-1, SGC790	[61,62]
		Liver cancer	HepG2, Hep3B	[64]
	↑ Cytochrome c	Oral cancer	HSC3	[57]
		Melanoma	M17, SP6.5	[58]
	↑ ROS	Colorectal cancer	SW-480	[65]
		Non-small lung cancer	NCI-H460	[59]
		Gastric cancer	AGS, SNU-1	[61]
	↓ PI3K, Akt	Pancreatic cancer	PANC-1	[63]
		Breast cancer	4T1, MDA-MB-453	[60,66]
Morin	↓ Bcl2 ↑ Bax	Myeloid leukemia	K562, KCL22	[67]
		Colorectal cancer	HCT-116. SW480	[68,69]
		Melanoma	G361, SK-MEL-2	[70]
	↑ caspases activation	Myeloid leukemia	K562, KCL22	[67]
		Colorectal cancer	HCT-116, SW480	[68,69]
		Melanoma	G361, SK-MEL-2	[70]
	↓ PI3K, Akt	Myeloid leukemia	K562, KCL22	[67]
	↑ Cytochrome c	Colorectal cancer	HCT-116	[68]
	↑ ROS	Melanoma	G361, SK-MEL-2	[70]
		Colorectal cancer	SW480	[69]
	↑ FAS	Colorectal cancer	HCT-116	[68]

**Table 3 nutrients-16-01201-t003:** Summary of the most important autophagy induction pathways involving flavonols and their impact on individual types of cancer. (↑-increase,↓ decrease).

Flavonoid	Mechanism	Cancer Type	Cell Line	Ref.
Quercetin	↓ Akt-mTOR	Breast cancer	MCF-7, MDA-MB-231	[97]
		Liver cancer	SMMC7721, HepG2	[98]
		Acute myeloid leukemia	HL-60	[99]
	↑ LC3	Liver cancer	SMMC7721, HepG2	[98]
		Neuroglioma	U87	[100]
		Acute myeloid leukemia	HL-60	[99]
		Lung cancer	A549, H1299	[101]
	↑ Beclin 1	Neuroglioma	U87	[100]
		Lung cancer	A549, H1299	[101]
	↑ Atg5, Atg7, Atg12	Lung cancer	A549, H1299	[101]
Kaempferol	↓ Akt-mTOR	Lung cancer	A549, H1299	[102]
		Liver cancer	SK-HEP-1	[103]
	↑ LC3	Lung cancer	A549, H1299	[102]
		Liver cancer	SK-HEP-1	[103]
		Oral cancer	MC-3	[104]
		Gastric cancer	SNU-638	[105]
		Ovarian cancer	A2780	[106]
		Prostate cancer	PC-3	[107]
	↑ Beclin 1	Lung cancer	A549, H1299	[102]
		Liver cancer	SK-HEP-1	[103]
		Oral cancer	MC-3	[104]
		Gastric cancer	SNU-638	[105]
		Ovarian cancer	A2780	[106]
	↑ Atg5, Atg7, Atg12	Liver cancer	SK-HEP-1	[103]
		Gastric cancer	SNU-638	[105]
		Ovarian cancer	A2780	[106]
Galangin	↓ Akt-mTOR	Laryngeal cancer	TU212, HEP-2	[108]
	↑ LC3	Laryngeal cancer	TU212, HEP-2	[108]
		Liver cancer	HepG2	[109,110]
	↑ Beclin 1	Laryngeal cancer	TU212, HEP-2	[108]
		Liver cancer	HepG2	[109,110]
Myricetin	↓ Akt-mTOR	Gastric cancer	AGS	[42]
		Colorectal cancer	HCT116, SW620	[47]
		Liver cancer	HepG2	[111]
	↑ LC3	Gastric cancer	AGS	[42]
		Colorectal cancer	HCT116, SW620	[47]
		Liver cancer	HepG2, Hep3B	[111,112,113]
	↑ Beclin 1	Gastric cancer	AGS	[42]
Isorhamnetin	↓ Akt-mTOR	Gastric cancer	MKN-45	[54]
Fisetin	↓ Akt-mTOR	Prostate cancer	PC3	[114]
	↑ LC3	Prostate cancer	PC3	[114]
		Oral cancer	Ca9-22	[115]
		Pancreatic cancer	PANC-1, BxPC-3	[116]
	↑ Beclin 1	Oral cancer	Ca9-22	[115]
	↑ Atg5	Oral cancer	Ca9-22	[115]

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
