# Peer review of "Selected Flavonols Targeting Cell Death Pathways in Cancer Therapy: The Latest Achievements in Research on Apoptosis, Autophagy, Necroptosis, Pyroptosis, Ferroptosis, and Cuproptosis"

_nutrients, 2024, doi:10.3390/nu16081201_

Round 1
Reviewer 1 Report
Comments and Suggestions for Authors
The authors of this work reviewed a few flavonols that have been shown to have potential in the treatment of cancer. They talked about how these flavonols cause cell death in certain cancer cells by exhibiting mechanisms that cause cell death. The authors also came to the conclusion that because flavonols have a variety of effects on cancer cells, including them in cancer treatment plans offers potential advantages. While the discussion is relatively weak in terms of conclusions and future applications, the content is generally complete. I believe that strengthening is essential. Furthermore, it would be preferable if each mechanism of cell death could display a scheme figure. For the benefit of the authors, a few remarks are attached below.
1. Although flavonols come in a variety of forms, the authors have chosen seven particular compounds for further discussion as the primary flavonoid types. Nevertheless, how this filtering condition is established is beyond my understanding. One common flavonoid that has been shown to have important toxicological effects on cancer cells (apoptosis, autophagy, etc.) is myricetin; however, the authors did not mention it in the discussion. To be more objective, I believe the justifications for include these substances in the screening should be made explicit.
2. Fig. 1: Readers can compare and contrast different cell death mechanisms quickly and easily with the aid of this diagram. Nevertheless, cuproptosis was mentioned by the writers in the conclusion and title, but it wasn't included in this figure. Maybe this mechanism should be added to make things more complete.
3. Fig. 4: Different flavonols are not positioned on distinct paths or targets as compared to the preceding figures.
4. Cell lines are used as the reference evidence in each table. And why do the authors give up on presenting this data when we all know that the level of evidence from animal models or even clinical trials is higher?
5. While the discussion's content is comprehensive, the specific regulatory mechanism appears to be lacking in strength. As an illustration, a specific flavonol can cause five different ways for cells to die. Nevertheless, it makes sense that these five methods of cell death might not happen simultaneously. If the author could include a brief explanation of this topic, such as the selectivity or variations in the regulation of different cancer types or cell death mechanisms by particular flavonols, I believe it would have more significance for the readers.
6. The conclusion is not very strong. It would be beneficial to include at least a few viewpoints that highlight the potential uses and future research paths of these flavonols in controlling various cell death processes.
Comments on the Quality of English LanguageOnly minor editing of English language required.
Author Response
Re: Review nutrients-2945399
"Selected flavonols targeting cell death pathways in cancer therapy: the latest achievements in research on apoptosis, autophagy necroptosis, pyroptosis, ferroptosis and cuproptosis”
Dominika Wendlocha, Robert Kubina, Kamil Krzykawski and Aleksandra Mielczarek-Palacz
Manuscript Status: Pending major revisions
The authors greatly appreciate valuable and positive comments provided by Reviewers which enhanced our manuscript. We have submitted a revised version of our manuscript “Selected flavonols targeting cell death pathways in cancer therapy: the latest achievements in research on apoptosis, autophagy necroptosis, pyroptosis, ferroptosis and cuproptosis". In order to prepare a corrected version, we have followed the comments made by the reviewers. We hope that our revisions will be found satisfactory and the manuscript will be suitable for publication. Please find our answers and acknowledgements to the Reviewers attached below.
Point-by-point answers to the Reviewer 1:
The authors of this work reviewed a few flavonols that have been shown to have potential in the treatment of cancer. They talked about how these flavonols cause cell death in certain cancer cells by exhibiting mechanisms that cause cell death. The authors also came to the conclusion that because flavonols have a variety of effects on cancer cells, including them in cancer treatment plans offers potential advantages. While the discussion is relatively weak in terms of conclusions and future applications, the content is generally complete. I believe that strengthening is essential. Furthermore, it would be preferable if each mechanism of cell death could display a scheme figure. For the benefit of the authors, a few remarks are attached below.
- Although flavonols come in a variety of forms, the authors have chosen seven particular compounds for further discussion as the primary flavonoid types. Nevertheless, how this filtering condition is established is beyond my understanding. One common flavonoid that has been shown to have important toxicological effects on cancer cells (apoptosis, autophagy, etc.) is myricetin; however, the authors did not mention it in the discussion. To be more objective, I believe the justifications for include these substances in the screening should be made explicit.
Reply: The selection of seven specific flavonols belonging to the flavonoid group for further discussion was made based on various factors such as the availability of scientific research, their potential biological significance, and relevance in a therapeutic context. Despite the wide variety of flavonol forms, we chose these seven compounds as representative of the flavonol group that are particularly studied and show promising potential in cancer research. One of the reasons for selecting specific compounds was the promising results of studies regarding their impact on cell death mechanisms in cancer cells. Additionally, they were chosen based on their potential influence on other significant biological processes such as inflammation, angiogenesis, or metastasis, which are crucial for cancer development. The work concerns the impact of flavonols on various types of cell death, so we selected compounds from the known flavonols, not the entire group of flavonoids. We described the effect of myricetin based on available literature for three types of cell death: apoptosis, autophagy, and pyroptosis. However, it is worth emphasizing that our selection of seven specific compounds does not exclude the possibility of considering other flavonoids in future research. Our ongoing research efforts will involve careful considerations regarding the selection of compounds for analysis to ensure the most comprehensive understanding of their impact on cancer processes.
- Fig 1: Readers can compare and contrast different cell death mechanisms quickly and easily with the aid of this diagram. Nevertheless, cuproptosis was mentioned by the writers in the conclusion and title, but it wasn't included in this figure. Maybe this mechanism should be added to make things more complete.
Reply: Thank you for your feedback. We have taken your suggestion into consideration and made revisions accordingly. In response to your comment, we have reorganized Figure 1 and included an additional type of cell death, namely cuproptosis. Furthermore, we have added an extra figure that incorporates the mechanisms of ferroptosis and necroptosis, as suggested in the introduction. We believe these changes will enhance the completeness of our presentation. Your input has been invaluable in improving the quality of our work..
- Fig 4: Different flavonols are not positioned on distinct paths or targets as compared to the preceding figures.
Reply: Thank you for your observation. We have revised the figures, and in accordance with your suggestion, we have standardized their format. As a result, the positioning of different flavonols in Figure 4 has been adjusted to align more closely with the preceding figures. We believe this change will enhance the clarity and consistency of the presentation. Your feedback has been instrumental in improving the quality of our work.
- Cell lines are used as the reference evidence in each table. And why do the authors give up on presenting this data when we all know that the level of evidence from animal models or even clinical trials is higher?
Reply: Thank you for your feedback. We acknowledge your point regarding the use of cell lines as reference evidence in each table. We appreciate your suggestion to include data from animal models or clinical trials, which typically provide higher levels of evidence. In response to your comment, we have prepared an entire separate chapter focusing specifically on in vivo studies. This chapter includes data obtained from animal models and, where available, clinical trials. We believe that by incorporating this additional information, we can provide a more comprehensive and robust analysis of the efficacy and potential therapeutic implications of the flavonoids under investigation. Your input has been invaluable in guiding us toward a more comprehensive presentation of our research findings. If you have any further suggestions or comments, please do not hesitate to let us know. We are committed to addressing all concerns and improving the quality of our work.
- While the discussion's content is comprehensive, the specific regulatory mechanism appears to be lacking in strength. As an illustration, a specific flavonol can cause five different ways for cells to die. Nevertheless, it makes sense that these five methods of cell death might not happen simultaneously. If the author could include a brief explanation of this topic, such as the selectivity or variations in the regulation of different cancer types or cell death mechanisms by particular flavonols, I believe it would have more significance for the readers.
Reply :We have added a brief explanation to the final chapter: conclusion
- The conclusion is not very strong. It would be beneficial these flavonols in controlling various cell death to include at least a few viewpoints that highlight the potential uses and future research paths of processes.
Reply: Thank you for your valuable insight. in accordance with the suggestion, we have included the conclusions at the end of the paper.
All changes suggested by the Reviewer were introduced to the corrected version of the text.
Authors made the corrections based on best understanding of P.T. Reviewers recommendations and we do express a sincere hope that our effort fullfilled entirely Reviewers suggestions.
Kind regards
Robert Kubina and co-authors
Reviewer 2 Report
Comments and Suggestions for Authors
The article entitled Selected flavonols targeting cell death pathways in cancer therapy: the latest achievements in research on apoptosis, autophagy necroptosis, pyroptosis, ferroptosis, and cuproptosis has excellent writing quality and provides a rich and current review of the main flavonoids in the fight against different types of cancer, presenting the different forms of action and the routes through which the fight against cancer cells occurs considering different compounds of this class. The Tables and figures are well-elaborated and pertinent to the manuscript's understanding. Despite this, the article needs to be revised in terms of formatting. At some points, the authors should have remembered to space between words and to place the period in the correct position (after the citation). In conclusion, the article is of great value to the area and will be cited extensively.
Author Response
Re: Review nutrients-2945399
"Selected flavonols targeting cell death pathways in cancer therapy: the latest achievements in research on apoptosis, autophagy necroptosis, pyroptosis, ferroptosis and cuproptosis”
Dominika Wendlocha, Robert Kubina, Kamil Krzykawski and Aleksandra Mielczarek-Palacz
Manuscript Status: Pending major revisions
The authors greatly appreciate valuable and positive comments provided by Reviewers which enhanced our manuscript. We have submitted a revised version of our manuscript “Selected flavonols targeting cell death pathways in cancer therapy: the latest achievements in research on apoptosis, autophagy necroptosis, pyroptosis, ferroptosis and cuproptosis". In order to prepare a corrected version, we have followed the comments made by the reviewers. We hope that our revisions will be found satisfactory and the manuscript will be suitable for publication. Please find our answers and acknowledgements to the Reviewers attached below.
Reply: Thank you for the review. We have conducted a thorough revision of the text and corrected all irregularities.
All changes suggested by the Reviewer were introduced to the corrected version of the text.
Authors made the corrections based on best understanding of P.T. Reviewers recommendations and we do express a sincere hope that our effort fullfilled entirely Reviewers suggestions.
Kind regards
Robert Kubina and co-authors
Round 2
Reviewer 1 Report
Comments and Suggestions for Authors
Reviewer thanks the authors for their replies. The authors have responded appropriately to all questions I raised.